# Copper-Electroplating-Modified Liquid Metal Microfluidic Electrodes

**DOI:** 10.3390/s22051820

**Published:** 2022-02-25

**Authors:** Jiahao Gong, Bingxin Liu, Pan Zhang, Huimin Zhang, Lin Gui

**Affiliations:** 1Liquid Metal and Cryogenic Biomedical Research Center, Technical Institute of Physics and Chemistry, Chinese Academy of Sciences, 29 Zhongguancun East Road, Haidian District, Beijing 100190, China; gongjiahao18@mails.ucas.ac.cn (J.G.); liubingxin17@mails.ucas.ac.cn (B.L.); zhangpan18@mails.ucas.ac.cn (P.Z.); zhanghuimin19@mails.ucas.ac.cn (H.Z.); 2CAS Key Laboratory of Bio-Inspired Materials and Interfacial Science, Technical Institute of Physics and Chemistry, Chinese Academy of Sciences, 29 Zhongguancun East Road, Haidian District, Beijing 100190, China; 3University of Chinese Academy of Sciences, 19 Yuquan Road, Shijingshan District, Beijing 100039, China

**Keywords:** electroplating, liquid metal microelectrodes, capacitive sensing

## Abstract

Here, we report a novel technology for the fabrication of copper-electroplating-modified liquid metal microelectrodes. This technology overcomes the complexity of the traditional fabrication of sidewall solid metal electrodes and successfully fabricates a pair of tiny stable solid-contact microelectrodes on both sidewalls of a microchannel. Meanwhile, this technology also addresses the instability of liquid metal electrodes when directly contacted with sample solutions. The fabrication of this microelectrode depends on controllable microelectroplating of copper onto the gallium electrode by designing a microelectrolyte cell in a microfluidic chip. Using this technology, we successfully fabricate various microelectrodes with different microspacings (from 10 μm to 40 μm), which were effectively used for capacitive sensing, including droplet detection and oil particle counting.

## 1. Introduction

Metal microelectrodes are essential parts of microfluidic systems. Microfluidic chips integrated with microelectrodes can be used for sample detection [1,2], drug delivery [3,4,5], cell screening [6,7,8], droplet manipulation [9,10], and microfluidic fuel cells [11,12]. The fabrication of metal microelectrodes depends on a variety of microfabrication technologies. The selection of appropriate microfabrication processes can meet the shape, size, and sensitivity requirements in microelectrode manufacturing and related application fields. Among them, sidewall microelectrodes [13,14,15] are widely used in microfluidic systems. They can produce uniform lateral electric fields along the vertical direction in the channel, which is commonly used for electrochemical sensing [16,17,18], electrical impedance sensing [19,20], capacitive sensing [21,22,23], among other applications [24].

Sidewall microelectrodes are usually fabricated by patterning the metal film on a flat substrate prior to sealing. There are several methods to fabricate metal thin-film electrodes on sidewalls. One of the approaches is to use shadow masks to pattern metal films at a certain angle. Kang et al. [15] utilized shadow effects occurring during angled evaporation through a shadow mask to deposit the metal films on sidewalls. Choi et al. [25] patterned 3D sidewall electrodes by ion implantation at a 40° angle with a metal shadow mask. This approach is complex and requires precise alignment technology, resulting in poor sealing. Chemical deposition can also be used to fabricate sidewall metal thin-film electrodes. Kadilak et al. [14] presented an approach to selectively deposit chemically bonded gold electrodes onto PDMS microchannel sidewalls. However, this approach requires multistep chemical reactions in the microchannel, with high cost. Other approaches to fabricate sidewall electrodes include electrodeposition [26] and ion milling [27]. They both require specialized equipment and complicated processes.

Injection of low melting point metals or alloys in microchannels is a cheap and effective way to fabricate microelectrodes [28,29,30] because it avoids multistep fabrication and complex alignment processes. The microelectrode shape is only determined by the shape of the microchannel through lithography. Liquid metals such as gallium and gallium-based alloys are commonly used injectable metals. Dickey et al. [30] utilized eutectic gallium indium (EGaIn) to fabricate a sidewall microelectrode in direct contact with a microfluidic channel. This sidewall microelectrode is easy to fabricate and flexible, it does not have an alignment process, and it enhances the intensity of the applied electric field, which is more effective for sorting, mixing, and amplifying electric signals for sensing.

However, the EGaIn liquid metal contact electrode has poor stability because it is highly prone to deformation when contacting fluids with pressure, resulting in electrode damage. In addition, liquid metal surfaces subjected to acid or alkali cause chemical corrosion, creating defects in the electrode. Instead of using EGaln, using pure gallium or bismuth-based alloys [31,32,33] with higher melting points can effectively resolve electrode deformation due to pressure because they are solid electrodes at room temperature. However, they are also intolerant of acid and alkali. Another limitation is that none of the above-mentioned electrodes can be fabricated to arbitrarily reduce the spacing between them. Thus, the detection sensitivity of the electrodes cannot be further improved. Therefore, we propose that if an inert metal with a high melting point is coated on the liquid metal’s surface, not only can the electrodes’ stability be increased, but their shapes and space can be further modified.

Microelectroplating [34,35,36,37] is a standard method to prepare or modify microelectrodes. It is a simple processing technology that can fabricate metal microelectrodes with more complex shapes and higher accuracy through a controllable electrochemical process. Bao et al. [35] proposed a method of electroplating Ag/AgCl to modify the microelectrode array for application in neural prosthesis. The prepared Ag/AgCl microelectrode has a higher charge storage capacity than the unmodified microelectrode, and the impedance decreases significantly. Polk et al. [36] utilized microelectroplating of silver on sharp edges to occlude large (103 mm^2^) vias in silicon substrates and leave open areas less than 1 μm^2^. The formation of solid-state micropores by this method is advantageous in microfluidics systems. However, the modification of a liquid metal microelectrode by microelectroplating technology on a microfluidic chip has not been reported before.

According to the fabrication of liquid metal contact electrodes and microelectroplating technology, this paper proposes a microfabrication technology for copper plating on sidewall gallium microelectrodes by designing a microelectrolyte cell on a microfluidic chip. This technology combines the advantages of easy fabrication of liquid metal electrodes and copper’s stable properties. The single-layer pair electrode structure evades complex alignment processes. In addition, since the plating process is performed after chip fabrication and packaging, the desired electrode-to-electrode spacing within 10–40 μm can be achieved by controlling the copper plating thickness, resulting in a suitable range of electrical signals to satisfy different detection requirements. Finally, this electroplated microelectrode pair is used for droplet capacitance detection and metal particle counting in oil. The results show that it can detect the ΔC ranging from 11 fF to 17 fF when the droplet length is within 68–248 μm and the ΔC ranging from 0.9 fF to 1.5 fF when the oil particle size is approximately within 8–15 μm. It suggests that microspacing sidewall microelectrodes based on liquid metal electroplating has great potential in particle detection.

## 2. Experimental Details

### 2.1. Design of the Chip

The design of a sidewall microelectroplated chip is based on the principle of an electrolytic cell. The chip consists of two polydimethylsiloxane elastomer (PDMS) slabs (2.5 cm long, 1.5 cm wide and 2.5 mm high), as shown in Figure 1a. Figure 1b shows the upper-layer structure of the chip. It consists of a transverse and two V-shaped electrode microchannels, both 50 μm high. The transverse microchannel includes an electrolyte inlet, an outlet, and a circular area (1.3 mm in diameter). The center of its circular area has a round hole (1.2 mm in diameter) used as a copper wire jack. Two V-shaped electrode microchannels are symmetrically distributed on both sides of the transverse flow channel. The microstructure in the V-shaped electrode microchannel is shown in the enlarged part of Figure 1b. A trapezoidal microgap (upper base: 28 μm wide) is set at the bottom of the electrode channel to avoid overflow of the liquid metal electrode. A rectangular microchannel (50 μm long and 28 μm wide) lies below the trapezoidal gap as an electroplating zone. After electroplating, it is connected with the transverse channel to form the contact electrode. The lower PDMS slab works as an encapsulation layer without any structure.

### 2.2. Fabrication of the Chip

The sidewall microelectroplated chip was fabricated using standard soft lithography. First, a photoresist SU-8 2050 (MicroChem Corp., Westborough, MA, USA) was used to mold a microchannel with the height of 50 μm on a silicon wafer (Ultrapak 100 mm, Fengcheng, Kaihua, China ). Second, the PDMS (a mixture of a base and a curing agent at a ratio of 10:1 by weight, Dow Corning, Midland, MI, USA) was cast on the silicon wafer and then baked at 75 °C for 1.5 h. After that, the microchannel pattern could be transferred to the PDMS surface. Third, the PDMS was cut into cuboid slabs (2.5 cm × 1.5 cm × 2.5 mm) containing the entire microchannel and peeled off from the silicon wafer. Fourth, another PDMS slab without any structures (2.5 cm × 1.5 cm × 2 mm) was used for plasma bonding with the upper structural layer to seal the microchannel (plasma cleaner, YZD08-2C, Tangshan Yanzhao Technology, Tangshan, China). Gallium (melting point: 29.8 °C; Shanxi Zhaofeng Gallium Co., Ltd., Quanyang, China) was firstly heated to liquid. The quantitative volume of liquid metal gallium was extracted by syringe and injected into a V-type electrode microchannel. Then, four wires (10 cm long and 0.2 mm diameter) were inserted into four holes of microelectrode channels, respectively. Another copper wire (1.2 mm diameter) was inserted into the copper wire slot. A 705 glue was used to seal all the wires. After the 705 glue curing, the sidewall microelectroplated chip was complete (the process is also shown in Appendix A). Finally, the chip was stored in a refrigerator at −80 °C to solidify the liquid metal gallium.

### 2.3. Method of Microelectroplating

The microchip was taken out from the refrigerator and observed under a microscope (Axio Observer Z1, Carl Zeiss, Oberkochen, German). A multimeter was used to check the conductivity of the electrode. The microelectroplating process is shown in Figure 2a. A four-channel microfluidic control system (MFCSTM-EZ, FLUIGENT, Paris, France) was used to pump the CuSO_4_ solution into the transverse microchannel. The pumping pressure was maintained at 300 mbar. After draining all the microchannel bubbles, CuSO_4_ solution could fill the entire electroplating region and fully contact the gallium surface. Next, the positive pole of a DC power supply (IT6720, 60 V/5 A/100 W, ITECH, Taiwan, China) was connected with the copper wire, and the negative pole was connected with the wire on one side of the gallium electrode. Based on the principle of an electrolytic cell, copper worked as a sacrificial metal that lost electrons and transformed into Cu^2+^ to enter the electrolyte. Secondly, the gallium cathode was continuously supplied with electrons through impressed current cathodic protection, inhibiting a self-corrosion reaction. Subsequently, Cu^2+^ migrated to the gallium surface to capture electrons to form Cu and deposited on the gallium surface, forming the copper coating. The copper layer continuously grew until the entire electroplating area was filled before stopping the process. The other side of the gallium electrode was electroplated in the same way. After electroplating on both sides of the electrode, DI water was continuously pumped into the electrolyte microchannel for 20 min to wash off the residual CuSO_4_ solution. Finally, the sidewall microelectroplating electrode pair was prepared with spacing of only 40 μm. (Figure 2b shows the actual diagram of the process). 

## 3. Results and Discussion

### 3.1. Effects of Electroless Plating

In the nonelectroplating process, when the gallium surface contacts the CuSO_4_ solution directly at room temperature, the oxidation–reduction reaction between Ga and Cu^2+^ can also occur, forming electroless plating. In addition, the electroless plating degree becomes more significant by increasing the concentration of CuSO_4_ solution, which seriously affects electroplating in this work because each electrode was electroplated separately. Consequently, when one of the electrodes was electroplated, the opposite electrode also contacted CuSO_4_ solution, forming the electroless plating, which resulted in the uncontrolled spontaneous growth of the copper layer. The effect of electroless plating on electroplating is shown in Figure 3a. However, the experimental results show that the low concentration of CuSO_4_ solution resulted in very weak electroless plating. Since the electroplating efficiency was much higher than that of electroless plating, the hazards of uncontrolled electroless plating could be minimized. Figure 3b,c show the microelectrodes’ morphology and electroless plating area curve over 20 min at different CuSO_4_ concentrations, respectively. The pump-in pressure of the CuSO_4_ solution was set at 300 mbar. When the concentration of CuSO_4_ was only 10%, the copper layer growth area was only about 130 μm^2^ in 20 min. However, with the CuSO_4_ concentration increasing to 20%, the copper layer area could reach about 1052 μm^2^ in 20 min. Therefore, choosing 10% CuSO_4_ as an electroplating solution was advantageous to eliminate the influence of electroless plating.

### 3.2. Voltage Effects on Microelectroplating

When the CuSO_4_ concentration is constant, the relationship between copper plating thickness, current density, and plating time is followed by the formula: L = K·D_K_·t·η_K_(1)
where L is plating thickness (μm), K is thickness coefficient, D_K_ is cathode current density, t is plating time, and η_K_ is cathode current density efficiency. The thickness coefficient K is determined by plating metal chemical equivalent α and plating metal density γ:K = α/γ(2)

This work calculates K as 0.133 cm^3^/ah according to a divalent copper ion chemical equivalent of 1.186 g/ah and a copper density of 8.9 g/cm^3^.

Formula (1) suggests that the plating thickness has a linear relationship with the plating time and that the electrodeposition rate V (=K·D_K_·η_K_) increases with the increase of current density. However, the required current density is minimal since this work was microplating at the microchannel with an extremely tiny plating area (50 μm × 28 μm). Therefore, the current delivered by the DC power supply cannot be reflected. However, the voltage provided by the DC power supply is proportional to the output current due to the specific internal resistance. Therefore, the plating thickness is also proportional to the voltage.

The copper layer growth of microelectrodes at different voltages was recorded by photography (shown in Appendix A). Figure 4a shows the thickness variation of the copper layer with time on one of the electrodes at 0.6 V, and the time required for the copper layer to fill the electroplating zone was 540 s. Figure 4b shows the thickness variation of the copper layer with time at different applied voltages. The copper layer thickness increased approximately linearly under the different plating voltages of 0.6, 0.8, and 1.0 V. The times required for the copper layer thicknesses to reach the maximum value were 540 s, 390 s, and 240 s, respectively. Their electroplating rates were calculated to be 0.13 μm/s, 0.18 μm/s, and 0.31 μm/s, respectively.

Figure 4c shows the average electroplating rate curve of the copper layer at different electroplating voltages (0.4, 0.6, 0.8, and 1.0 V). The copper plating rate increased with the increase of voltage. The average electroplating rate at 0.4, 0.6, 0.8, and 1.0 V was 0.06 μm/s, 0.13 μm/s, 0.18 μm/s, and 0.29 μm/s, respectively. However, suppose the voltage (>1 V) continued to increase. In that case, the plating rate might become less controllable. The copper layer might be less dense with the risk of copper dendrite generation. Consequently, in the actual microplating process, the ideal plating voltage range was from 0.6 V to 1 V.

### 3.3. Microelectrode Stability

To verify whether the Cu-plated Ga microelectrode had better stability than other common liquid metal microelectrodes, we compared the Cu-plated Ga microelectrode with the EGaIn and Ga solid microelectrodes. The deformation degrees of these three kinds of electrodes were tested at different pressures or temperatures.

In the pressure resistance test, DI water (25 °C) was pumped into the transverse microchannel at different pressures (200 mbar, 250 mbar, 300 mbar, and 350 mbar) by using the four-channel microfluidic system, and a microscope was used to observe and record their electrode conditions over 30 s at different pressures. Then, the deformation areas were estimated based on pixel points of the images. The EGaIn electrode began to deform at 200 mbar at 25 °C, and the deformation area was about 7505 μm^2^, as shown in Figure 5a. No deformations occurred at this pressure interval for the Ga solid electrode or the Cu-plated Ga electrode, demonstrating that these solid electrodes have higher pressure resistance at room temperature. 

Furthermore, the Ga solid electrode and the Cu-plated Ga electrode were used for the temperature resistance test. DI water with different inlet temperatures (45 °C, 55 °C, 65 °C, 75 °C, and 85 °C) was pumped into the transverse microchannel at 300 mbar, and the microscope was used to observe and record the electrode conditions over 30 s. Figure 5b shows that the Ga solid electrode began to deform at 75 °C and the deformation area was about 1744 μm^2^ because the Ga surface began to melt, resulting in poor pressure resistance. In contrast, the Cu-plated Ga electrode exhibited better temperature resistance, with no electrode deformation from 45 °C to 85 °C, attributed to the Cu layer protection on the low melting point Ga. In this case, the DI water with high temperature and pressure could not directly contact the Ga surface to induce deformation. Therefore, the experiment shows that the Cu-plated Ga microelectrode has better performance in both pressure and temperature resistance.

### 3.4. Fabrication of Microelectrodes with Different Spacings

Electrodes of different microdistances can also be fabricated using this microelectroplating method to meet different detection accuracies. We successfully fabricated microelectrodes with the distances of 40 μm, 30 μm, 20 μm, and 10 μm, as shown in Figure 6. In the fabrication process, the microscope was used to observe the growth of the copper layer. For the distance of 40 μm, we needed to ensure that the coatings of the upper and lower electrodes filled the whole plating area. However, for the distances of 30 μm, 20 μm, and 10 μm, the coatings needed to grow into the microchannel. Taking the fabrication of electrodes with a distance of 30 μm as an example, the upper electrode was first electroplated with the voltage of 0.6 V. After filling the plating area, an extra 5 μm was expected to grow in the vertical direction. Therefore, the required growth location was calibrated in advance, and when the copper layer arrived here, the plating of the upper electrode was stopped. Next, the same electroplating method was used for the lower electrode. Finally, we acquired the electrodes with a gap of 30 μm between each.

## 4. Applications

### 4.1. Droplet Detection

Based on the microdistance structure of liquid metal microelectroplated sidewall electrodes, small capacitance changes can be detected, such as capacitance changes of droplets with different lengths. Therefore, we fabricated a droplet capacitive microsensor that contained the microdistance electrodes. This microsensor was constructed by adding a T-shaped microchannel to the previously introduced microelectroplating chip shown in Figure 7a. The T-shaped microchannel generated water-in-oil droplets (dimethyl silicone oil is the continuous phase, and DI water is the discrete phase). The corresponding capacitance signal was generated when droplets passed through the sensing area containing electroplated electrodes. Figure 7b shows the construction of an experimental instrument. First, a four-channel microfluidic control system was used to pump the electrolyte, and a DC power supply was used to apply voltages for electroplating. The microfluidic control system was used again to adjust the oil pressure and DI water to generate uniform water-in-oil droplets of different lengths after the electroplating. The microelectrodes were connected with an LCR meter (TH2817A, Precision LCR Meter, Shenzhen, China) to detect capacitance signals. The data were recorded in the computer’s LabVIEW program. The microscope was used to observe the optical results during the whole process.

The capacitance change caused by a droplet is due to permittivity variation between microelectrodes. The relative permittivity values of DI water and dimethyl silicone oil are 81 and 2.5, respectively. Therefore, when a water-in-oil droplet passes through the microelectrodes, the permittivity between the electrodes increases, thus increasing the capacitance value. In the experiment, droplets of lengths 68 μm, 133 μm, and 248 μm were generated by adjusting the pressure of oil and water phases. The LCR meter detected the capacitance values when they passed through the electrodes (electrode spacing: 40 μm). Figure 8a shows the detection results of 68 μm droplets over 30 s at 0.1 MHz. Each droplet could generate a capacitive signal, and the average capacitance change (ΔC) was 11 fF. However, the capacitive signals fluctuated significantly. This is because a tiny droplet passed through the electrodes. The low sampling frequency for the LCR meter resulted in a significant measurement error.

Nevertheless, for 133 μm droplets, the measurement results improved, as shown in Figure 8b. The average ΔC caused by 133 μm droplets over 30 s could reach 14 fF at 0.1 MHz, making the signals more stable. Figure 8c recorded the real-time capacitance changes of the whole process of a 248 μm droplet passing through the electrode. In the figure, a1 (t = 0.2 s) is the location before the droplet entered the electrode region, and a2, a3, a4, and a5 are the locations of the microelectrode region at t = 0.4 s, 0.6 s, 0.8 s, and 1.0 s, respectively. The capacitance reached the maximum at a3, where the ΔC was 17 fF. The droplet left the microelectrode region completely at a6. Thus, the capacitance returned to the initial value. The above results suggest that this microdistance microelectrode based on microelectroplating can count droplets with different lengths and record the capacitance response process of a long droplet.

### 4.2. Metal Particle Counting in Oil

Metal particle pollution in oil [38,39,40,41] is one of the major causes of mechanical failure. The material, size, and concentration of pollutants can reflect the working state of mechanical equipment. Therefore, metal particle counting in oil is significant to mechanical systems’ safety and fault diagnosis. Common metal contamination particles include aluminum, iron, and copper. Shi et al. [42] fabricated a microsensor to count Fe and Cu particles in oil by the inductance principle. However, their device was mainly used for detecting large metal particles (>30 μm). Murali et al. [43] fabricated a microfluidic counter for 10–25 μm aluminum capacitance detection in lubricating oil. However, the metal film electrodes were made by evaporation, which increased the chip fabrication cost and effort.

The microdistance microelectrodes based on liquid metal microelectroplating introduced in this paper are a better tool to detect smaller oil particles because the microdistance can improve the sensitivity of capacitance detection. The chip is also simple to fabricate. Consequently, we utilized this microelectrode to detect the capacitance of copper particles in the oil. First, a small number of copper particles (size: 5–20 μm) was mixed and dispersed in silicone oil. Then, the oil was pumped into the DC channel of the microchip after the electroplating process. The LCR meter was used to detect the capacitance signal at the passing of oil particles. Since metal particles’ permittivity approaches positive infinity at high frequency, the relative permittivity between the microelectrodes increases when a copper particle passes through the microelectrodes, increasing capacitance.

Optical and capacitance counting results of copper particles are shown in Figure 9a,b, respectively (real-time recording is shown in Appendix A). Eight copper particles (a–h) were recorded over 200 s through the electroplated microelectrodes. The distance between the microelectrodes was about 35 μm, and the particle size was approximately 8–15 μm with irregular particle shapes. We selected the peak values that were far higher than the noise values of the LCR meter as the basis for particle counting. The average noise value was calculated as 0.2 ± 0.1 fF. The ΔC caused by different particles (a–h) was within 0.9–1.5 fF at 0.1 MHz. However, for smaller copper particles (<8 μm), the resulting capacitive signals were covered by noise, making them unrecognizable. For comparison, we also utilized nonelectroplated liquid metal microelectrodes to detect copper particles in oil (Appendix A). However, due to excessive electrode spacing, no capacitance signal caused by copper particles could be detected by the LCR meter. It proved that this novel microdistance microelectrode based on microelectroplating has higher sensitivity to detect small oil particles.

For practical applications, we also believe that this highly sensitive capacitive sensing technology based on the microelectroplating process can monitor and evaluate biological or environmental samples of different components, sizes, and concentrations in real time. In biological applications, the biochip integrated with this capacitive sensor can achieve the collection, detection, and screening of biological droplets, such as sweat and urine, for health monitoring. For environmental applications, the sensor can monitor microparticle size and concentration of water or oil pollution in industries to evaluate the environmental indicators.

## 5. Conclusions

This paper proposes a novel technology for the fabrication of microdistance sidewall microelectrodes based on liquid metal microelectroplating. This technology overcomes the complicated fabrication of traditional sidewall solid electrodes in microfluidics. The liquid metal electrode modified by electroplating has better stability than ordinary liquid electrodes as a contact-type electrode. In addition, sidewall microelectrodes with different microdistances can be more easily fabricated by this microelectroplating method, which significantly increases the detection sensitivity of microelectrodes. In applications, the sidewall microelectrodes fabricated by this technology also proved excellent capacitance detection performance, detecting 68–248 μm microdroplets and 8–15 μm metal particles in the oil. Therefore, this novel method for microelectrode fabrication has definite value in manufacturing high-sensitivity microsensors in the future.

## Figures and Tables

**Figure 1 sensors-22-01820-f001:**
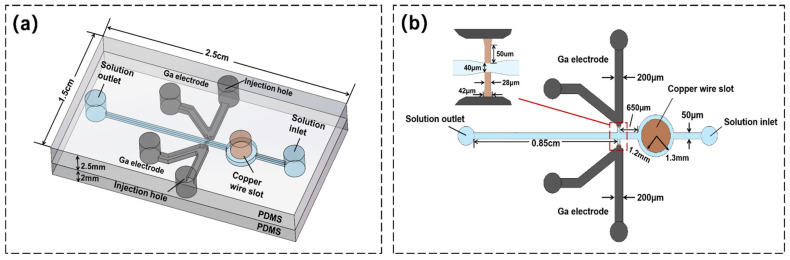
Chip structure diagram: (**a**) 3D structure diagram of the chip. (**b**) Plane diagrammatic sketch of the chip. The transverse electroplating solution microchannel is 50 μm wide. The electrode microchannel is 200 μm wide. The diameter of the circular microchannel is 1.3 mm, and the diameter of copper wire is 1.2 mm. The vertical distance from the circular channel to the electrode is 650 μm. For the microstructure part, the upper bottom of the trapezoidal gap is 42 μm. The lower bottom is 28 μm. The rectangular area is 28 μm wide and 50 μm long. The microchannel width between the electrode areas is 40 μm.

**Figure 2 sensors-22-01820-f002:**
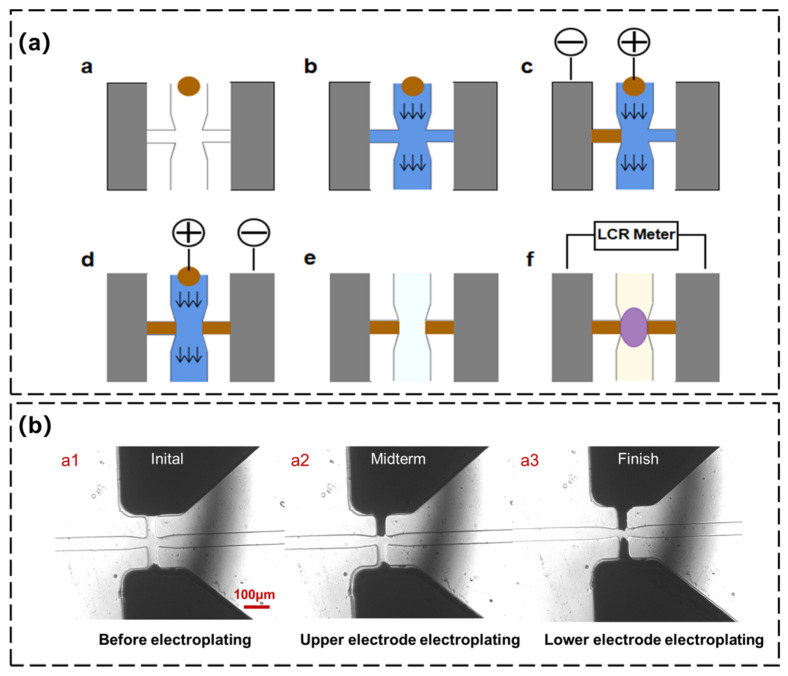
Microelectroplating process: (**a**) Microelectroplating steps: **a**. the initial state of the microelectrode; **b**. pumping CuSO_4_ solution; **c**. Ga electrode plating on one side; **d**. Ga electrode plating on the other side; **e**. DI water cleaning; **f**. applications after electroplating. (**b**) Actual diagram of electroplating process. Subfigures **a1**, **a2** and **a3** were plating conditions at different stages.

**Figure 3 sensors-22-01820-f003:**
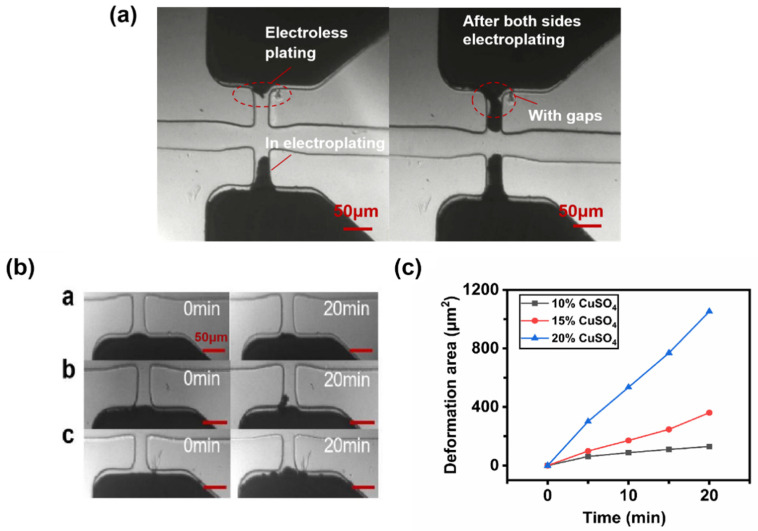
Effect of electroless plating: (**a**) Effect of electroless plating on the microelectroplating process by CuSO_4_ with high concentration. (**b**) Electroless plating conditions of microelectrodes over 20 min at different CuSO_4_ concentrations: **a**. 10% CuSO_4_; **b**. 15% CuSO_4_; **c**. 20% CuSO_4_. (**c**) Electroless plating areas at different CuSO_4_ concentrations.

**Figure 4 sensors-22-01820-f004:**
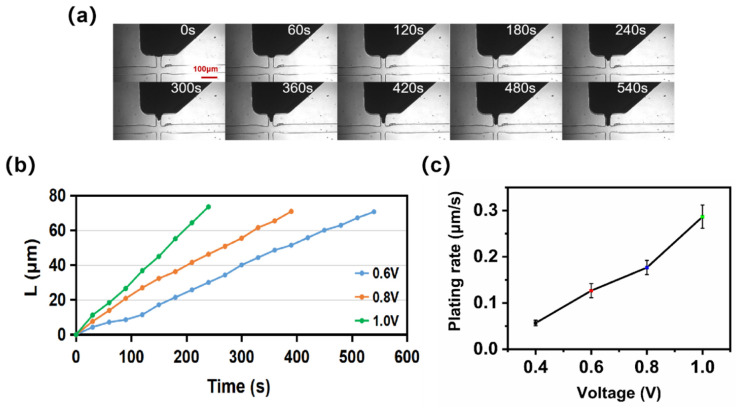
Effect of voltage on microplating: (**a**) Photograph of copper layer growth with time at 0.6 V. (**b**) The curve of copper layer thickness versus time at different voltages. (**c**) Average electroplating rate at different applied voltages.

**Figure 5 sensors-22-01820-f005:**
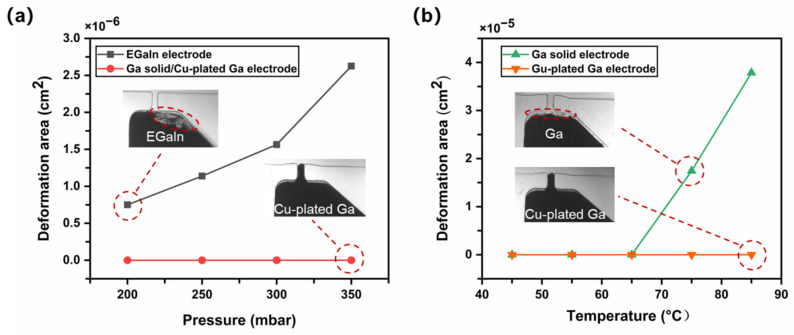
Pressure and temperature resistance tests of different microelectrodes: (**a**) Pressure resistance comparison between EGaIn and Ga solid/Cu-plated Ga electrodes at 25 °C. (**b**) Temperature resistance comparison between Ga solid electrode and Cu-plated Ga electrode at 300 mbar.

**Figure 6 sensors-22-01820-f006:**
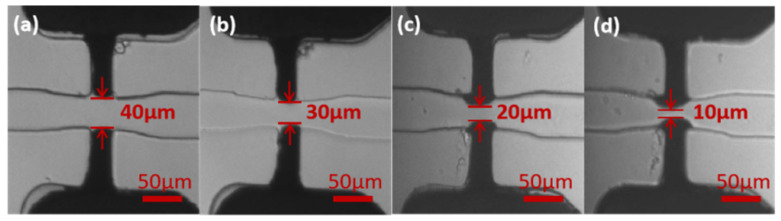
The fabrication of microelectrodes with different microdistances: (**a**) 40 μm spacing; (**b**) 30 μm spacing; (**c**) 20 μm spacing; (**d**) 10 μm spacing.

**Figure 7 sensors-22-01820-f007:**
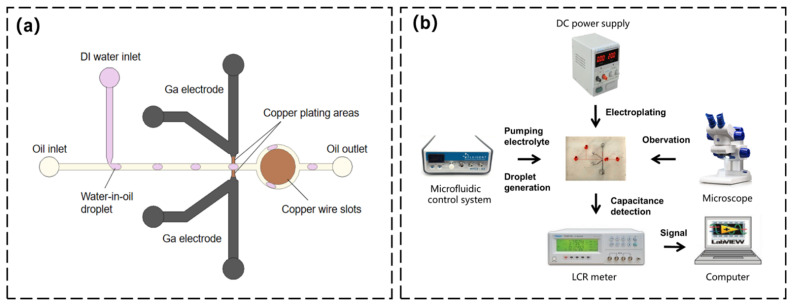
(**a**) The structure of a droplet detection sensor based on the microelectroplating electrode. (**b**) Diagram of the test setup.

**Figure 8 sensors-22-01820-f008:**
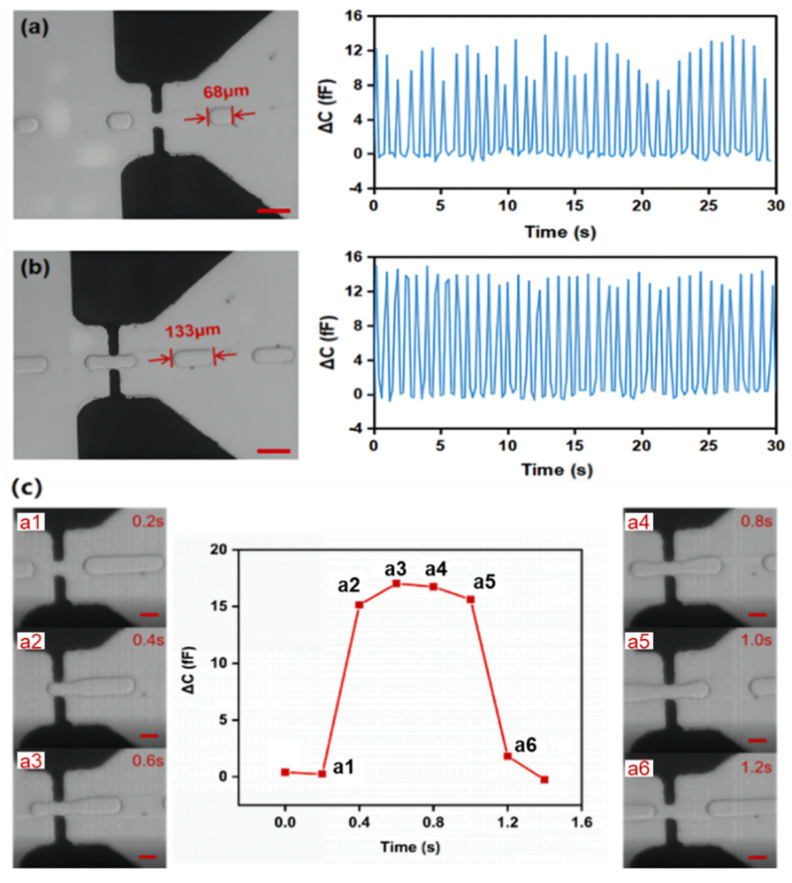
Droplet capacitance detection: (**a**) Capacitance signal caused by 68 μm droplets over 30 s. Scale bar, 100 μm. (**b**) Capacitance caused by 133 μm droplets over 30 s. Scale bar, 100 μm. (**c**) Capacitance response process caused by a 248 μm droplet. Subfigures a1–a6 show the position of the droplet at different times. Scale bar, 50 μm.

**Figure 9 sensors-22-01820-f009:**
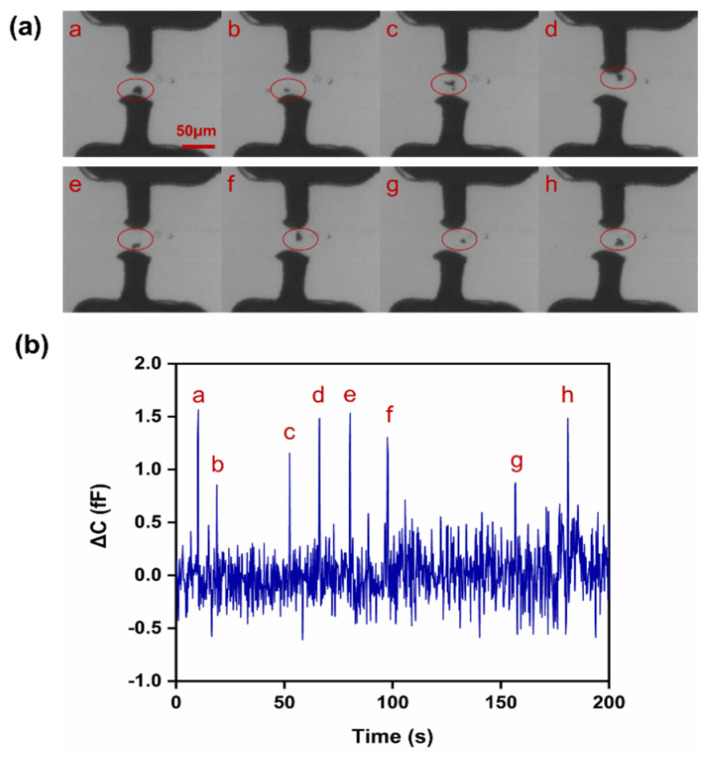
Metal particle counting in oil: (**a**) optical counting result of copper particles in oil over 200 s. Subfigures **a**–**h** indicate different copper particles passing through the electrodes, respectively; (**b**) the capacitance detection result of copper particles in oil over 200 s at 0.1 MHz.

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
