# Peer review of "Copper-Electroplating-Modified Liquid Metal Microfluidic Electrodes"

_sensors, 2022, doi:10.3390/s22051820_

Round 1

Reviewer 1 Report

This manuscript reports a fabrication approach for liquid metal-based microelectrodes. The micro- electroplating process was achieved based on the principle of an electrolytic cell, and a pair of sidewall electrodes was fabricated. The authors studied the effects of electroless plating, the applied voltage, and the stability of the electrodes, and demonstrated two applications of the fabricated liquid metal-based electrodes. The present study would be of interest to researchers in the fields of sensors, microfluidics and MEMS. I have some minor comments below:

1. Can the authors explain why the washing process takes a relatively long time (i.e. about 20 minutes)?

2. It seems that the concentration of CuSO4 solution has a significant effect on the electroplating. Can the authors provide an explanation?  

3. Does the plating rate increase with the increase of voltage? Why the ideal plating voltage is in the range from 0.6V to 1V?

4.  Can the authors provide more experimental details, e.g. how do you adjust the plating voltage and time to obtain different plating distance? How do you select the peaks for counting the number of cooper particles as shown in Figure 9?

5.  The authors are suggested to explain what the  stands for?  Also, the authors are suggested to comment on the stability and repeatability, as they impact the sensing performance.

Author Response

Response to reviewers’ comments:

This manuscript reports a fabrication approach for liquid metal-based microelectrodes. The micro- electroplating process was achieved based on the principle of an electrolytic cell, and a pair of sidewall electrodes were fabricated. The authors studied the effects of electroless plating, the applied voltage, and the stability of the electrodes, and demonstrated two applications of the fabricated liquid metal-based electrodes. The present study would be of interest to researchers in the fields of sensors, microfluidics and MEMS. I have some minor comments below:

  1. Can the authors explain why the washing process takes a relatively long time (i.e. about 20 minutes)?

Reply: Thanks for the reviewer’s comment. There are mainly two reasons why the washing process takes a relatively long time. First, the pumping pressure of water we set is only 100 mbar, the water did not flow fast in the microchannel due to relatively high flow resistance. It means that the cleaning speed is not fast, thus we should increase the cleaning time to ensure the effect of cleaning. Second, because there are some narrow gaps in the chips which is not easy to clean, we want to ensure that all residual CuSO4 in the microchannel is drained. The purpose we guarantee a long enough washing time is to prevent residual CuSO4 crystallization from blocking the microchannels after a period of time and also prevent CuSO4 from contaminating droplet and oil particle samples in later use.

  1. It seems that the concentration of CuSO4 solution has a significant effect on the electroplating. Can the authors provide an explanation?

Reply: Thanks for the reviewer’s comment. First, we want to explain the effect of electroless plating caused by CuSO4 concentration on electroplating in our work. The metal we used to be plated is Gallium. The metal activity order of Ga is greater than that of Cu, so a replacement reaction would happen between Ga and CuSO4 under natural conditions. If the concentration of CuSO4 increases, the chemical reaction rate would also increase leading to the chemical deposition of Cu on Ga, but this kind of chemical deposition is not uniform and dense (shown in the left picture of fig. 3(a)) due to Insufficient electroless plating of Cu on Ga (Ga reacts slowly with CuSO4, which was verified by our macro experiment). In this case, when we start the electroplating, Cu is more likely to grow along with the position of the previous electroless plating, causing the coating at the root of the electrode is not dense (shown in the right picture of fig. 3(a)). Therefore, we need to reduce the concentration of CuSO4 solution to maximize the elimination of electroless plating. Second, on the other hand, it is known that a higher concentration of salt solution leads to higher current efficiency, causing faster deposition of metal. Based on these two points, the concentration of CuSO4 solution has a significant effect.

  1. Does the plating rate increase with the increase of voltage? Why the ideal plating voltage is in the range from 0.6V to 1V?

Reply: Thanks for the reviewer’s comment. Yes, both the theory and the experiments we provide indicate that the plating rate increase with the increase of voltage. We choose the voltage range from 0.6V to 1V mainly for two reasons. First, If the voltage is lower than 0.6V, electroplating can also be achieved. Considering the substantial increase of the plating time (>10min) with too low voltage (<0.5V), it’s better to increase the voltage (≥0.6V) to improve the plating efficiency. Second, if the voltage is greater than 1V, we found that the thickness of the copper layer in the vertical direction increases faster, but sometimes the horizontal direction is too late to be fully covered with the copper layer, resulting in uneven plating with gaps. Besides, if the voltage is too high, excessive cathode current density would lead to the generation of dendrites. Therefore, the ideal plating voltage range is from 0.6V to 1V.

  1. Can the authors provide more experimental details, e.g. how do you adjust the plating voltage and time to obtain different plating distance? How do you select the peaks for counting the number of cooper particles as shown in Figure 9?

Reply: Thanks for the reviewer’s suggestion. For the first question, actually, the method we obtain different plating distances mainly relies on microscope observation. We have added more experimental details in chapter 3.4. The contents are as follows:

“In the fabrication process, the microscope was used to observe the growth of the copper layer. For the distance of 40 μm, we just need to ensure that the coatings of the upper and lower electrodes fill the whole plating area. But for the distance of 30 μm, 20 μm, and 10 μm, the coating needs to grow into the microchannel. Taking the fabrication of electrodes with a distance of 30 μm as an example, the upper electrode was first electroplated with the voltage of 0.6 V. After filling the plating area, an extra 5 μm should grow in the vertical direction. Therefore, the required growth location was calibrated in advance, and when the copper layer arrived here, the plating of the upper electrode was stopped. Then, the same electroplating method was used for the lower electrode. Finally, we acquired the electrodes with a gas of 30 μm from each other. “

For the second question, we selected the peaks that are far higher than the noise values of the LCR meter as the basis for particle counting. The average noise value was calculated as 0.2±0.1fF when no particle passed by. (We have added this detail into the manuscript). The minimum ΔC was 0.9 fF (>>0.2fF) when the particle passed by seen from the microscope. Therefore, the particles were easily counted by these significant peak signals.

  1. The authors are suggested to explain what the stands for?  Also, the authors are suggested to comment on the stability and repeatability, as they impact the sensing performance.

Reply: Thanks for the reviewer’s suggestion. For the first question, it seems that there is one word missing in the question “the ? stands for”, we have checked all the symbols again and added corresponding explanations. For the second question, we discuss this from two points: the electroplating process and sensing performance. For the electroplating process, according to the statistics of chip fabrications (more than 50 chips), the success rate of the electroplating process was about 70%, which proved the repeatability of this technology. some failures came mainly from the shedding of the plating layer due to the uneven plating surface. For the electrodes after successful electroplating, they could maintain good condition for more than at least 5 days if we filled the transverse microchannel with oil. Therefore, we could ensure the stability of the electrodes for a long time. For the sensing performance, the droplet sensing experiments last at least 2 hours and no damage was found after the experiments. The chip could be reused after appropriate storage.

Reviewer 2 Report

This study describes a microfluidic device for copper electroplating on liquid metal and its application for capacitive sensing of microparticles in the device. Although liquid metal previously was utilized as sensing electrodes in a microfluidic device, in the present study, Cu was electroplated on the liquid metal for improving the stiffness of the structures, which seems to be a novel point. This study is interesting, and this paper is well organized. After minor modifications (see comments), this reviewer would recommend acceptance.

Comments

  1. The tile is a little strange for this reviewer. As the microfluidics is the main topic, how about adding “microfluidic” into the title?
  2. If possible, please add a short comment on the practical applications. How is the capacitive sensing of droplets and microparticles used for practical applications?
  3. A space should be added between a number and a unit. For example, “10µm” should be changed to "10 µm".
  4. “4” of “CuSO4” should be written in subscript.
  5. As some spaces have been gone, the authors should add them. For example, “(a)Effect” should be changed to “(a) Effect”.
  6. “(a)Effect of Electrodeless Plating” might be changed to “(a) Effect of electrodeless plating”.
  7. There is no information on what a, b, and c indicate in Fig. 3(b).
  8. The authors might modify the writing of unit. For example, “0.4V, 0.6V, and 1.0V” might be changed to “0.4, 0.6, and 1.0 V”.
  9. There is no information on what “a1” and “a2” indicate in Fig. 8(c). Maybe, these images have been gone from Fig. 8(c).
  10. When “EGaIn” appears at the first time, the authors should use “eutectic gallium indium (EGaIn)”.
  11. “figure 1(a)” might be changed to “Figure 1(a)”.
  12. “2+” of “Cu2+” should be written in superscript.
  13. “50×28µm” might be changed to “50 µm × 28 µm”.

Author Response

Response to reviewers’ comments:

This study describes a microfluidic device for copper electroplating on liquid metal and its application for capacitive sensing of microparticles in the device. Although liquid metal previously was utilized as sensing electrodes in a microfluidic device, in the present study, Cu was electroplated on the liquid metal for improving the stiffness of the structures, which seems to be a novel point. This study is interesting, and this paper is well organized. After minor modifications (see comments), this reviewer would recommend acceptance.

Comments

  1. The tile is a little strange for this reviewer. As the microfluidics is the main topic, how about adding “microfluidic” into the title?

Reply: We highly agree with the reviewer’s suggestion. We have revised the title to “Copper electroplating modified liquid metal microfluidic electrodes.”

  1. If possible, please add a short comment on the practical applications. How is the capacitive sensing of droplets and microparticles used for practical applications?

Reply: Thanks for the reviewer’s suggestion. We have added a short comment about the practical applications at the end of chapter 4. The contents are as follows: “For practical applications, we believe that this highly sensitive capacitive sensing technology based on the micro-electroplating process can monitor and evaluate biological or environmental samples of different components, sizes, and concentrations in real-time. In biological applications, the biochip integrated with this capacitive sensor can achieve the collection, detection, and screening of biological droplets such as sweat and urine for health monitoring. For environmental applications, the sensor can monitor the microparticle size and concentration of water or oil pollution in industries to evaluate the environmental indicators.”

  1. A space should be added between a number and a unit. For example, “10µm” should be changed to "10 µm".

Reply: Thanks for the reviewer’s suggestion. We have checked the full text and added a space between each number and unit.

  1. “4” of “CuSO4” should be written in subscript.

Reply: Thanks for the reviewer’s suggestion. We have revised the “4” of “CuSO4” in our manuscript according to the reviewer’s suggestion.

  1. As some spaces have been gone, the authors should add them. For example, “(a)Effect” should be changed to “(a) Effect”.

Reply: Thanks for the reviewer’s comment. We have checked the full text again and revised these mistakes.

  1. “(a)Effect of Electrodeless Plating” might be changed to “(a) Effect of electrodeless plating”.

Reply: Thanks for the reviewer’s comment. We have revised this mistake in the manuscript.

  1. There is no information on what a, b, and c indicate in Fig. 3(b).

Reply: Thanks for the reviewer’s suggestion. We have added the information about a, b and c in fig. 3(b), which can be seen in the figure note.

  1. The authors might modify the writing of unit. For example, “0.4V, 0.6V, and 1.0V” might be changed to “0.4, 0.6, and 1.0 V”.

Reply: Thanks for the reviewer’s comment. We have revised this mistake in the manuscript.

  1. There is no information on what “a1” and “a8” indicate in Fig. 8(c). Maybe, these images have been gone from Fig. 8(c).

Reply: Thanks for the reviewer’s comment. a1 and a8 are the positions of the droplet at t=0s and t=1.4s, which were not indicated in fig. 8(c), because we think these two positions are unnecessary points to be described due to their far distance from the electrodes. In order to eliminate the misunderstanding, we have deleted the leftmost and rightmost marker point of a curve and changed the original “a2” to “a1”, so the following have been changed to “a2, a3, a4, a5, a6”. The modification can be seen in fig .8(c)

  1. When “EGaIn” appears at the first time, the authors should use “eutectic gallium indium (EGaIn)”.

Reply: Thanks for the reviewer’s comment. We have added this description to the manuscript according to the reviewer’s suggestion.

  1. “figure 1(a)” might be changed to “Figure 1(a)”.

Reply: Thanks for the reviewer’s comment. We have revised this mistake in the manuscript.

  1. “2+” of “Cu2+” should be written in superscript.

Reply: Thanks for the reviewer’s suggestion. We have revised this mistake in the manuscript.

  1. “50×28µm” might be changed to “50 µm × 28 µm”.

Reply: Thanks for the reviewer’s suggestion. We have revised this mistake in the manuscript.

Reviewer 3 Report

The authors state that this article proposes a new technology for manufacturing microelectrodes with side walls at the micro level based on microelectrolysis with liquid metal. They claim that this technology overcomes the complexity of traditional manufacturing of solid-state metal electrodes with side walls. They also claim that they successfully produces a pair of tiny stable microelectrodes with a solid contact on both side walls of the microchannel. They also claim to have successfully created a pair of tiny stable microelectrodes with a solid contact on both side walls of the microchannel. The authors also claim that the manufacture of a microelectrode depends on the controlled microelectrolysis of copper on a gallium electrode by creating a microelectrolyte cell in a microfluidic chip. Using this technology, the authors successfully manufactured various microelectrodes with different worldviews (from 10 microns to 40 microns), which were effectively used for capacitive sensing, including detection of droplets and counting of oil particles.

The article describes in detail the above described technological processes. Moreover, the authors claim that based on the micro-movement structure of the side electrodes with a micro electro coating of liquid metal, it is possible to detect small changes in capacitance, such as changes in the capacitance of droplets of different lengths. Therefore, they manufactured a drop capacitive microsensor containing micro-band electrodes. This microsensor was designed by adding a T-shaped microchannel to a previously introduced microelectronics chip.

Apparently, this new method of manufacturing microelectrodes has certain prospects in the production of highly sensitive microsensors in the future and the article undoubtedly deserves to be published in the journal Sensors.

Author Response

Response to the reviewer’s comments:

The authors state that this article proposes a new technology for manufacturing microelectrodes with side walls at the micro level based on microelectrolysis with liquid metal. They claim that this technology overcomes the complexity of traditional manufacturing of solid-state metal electrodes with side walls. They also claim that they successfully produces a pair of tiny stable microelectrodes with a solid contact on both side walls of the microchannel. They also claim to have successfully created a pair of tiny stable microelectrodes with a solid contact on both side walls of the microchannel. The authors also claim that the manufacture of a microelectrode depends on the controlled microelectrolysis of copper on a gallium electrode by creating a microelectrolyte cell in a microfluidic chip. Using this technology, the authors successfully manufactured various microelectrodes with different worldviews (from 10 microns to 40 microns), which were effectively used for capacitive sensing, including detection of droplets and counting of oil particles.

The article describes in detail the above described technological processes. Moreover, the authors claim that based on the micro-movement structure of the side electrodes with a micro electro coating of liquid metal, it is possible to detect small changes in capacitance, such as changes in the capacitance of droplets of different lengths. Therefore, they manufactured a drop capacitive microsensor containing micro-band electrodes. This microsensor was designed by adding a T-shaped microchannel to a previously introduced microelectronics chip.

Apparently, this new method of manufacturing microelectrodes has certain prospects in the production of highly sensitive microsensors in the future and the article undoubtedly deserves to be published in the journal Sensors.

Reply: Thank you very much for your affirmation and approval to our article.